# Somatic Characteristics and Special Motor Fitness of Young Top-Level Polish Table Tennis Players

**DOI:** 10.3390/ijerph18105279

**Published:** 2021-05-16

**Authors:** Beata Pluta, Szymon Galas, Magdalena Krzykała, Marcin Andrzejewski, Karolina Podciechowska

**Affiliations:** Faculty of Tourism and Recreation, Poznan University of Physical Education, Poland Królowej Jadwigi 27/39, 61-871 Poznan, Poland; galas.szymon@gmail.com (S.G.); krzykala@awf.poznan.pl (M.K.); andrzejewski@awf.poznan.pl (M.A.); podciechowska@awf.poznan.pl (K.P.)

**Keywords:** table tennis, young athletes, anthropometry, body composition, somatotype, special physical fitness

## Abstract

In the present study, we aimed to identify the impact of chosen anthropometric measurements on the special physical fitness of elite junior table tennis players at different stages of sport training. A total of 87 table tennis players aged 13.4 ± 1.74 years (43.7% girls and 56.3% boys) from two Polish teams were analyzed. The anthropometry measurements included height, sitting height, body weight, arm span, humerus and femur breadths, five skinfold thicknesses, and five girths were assessed. Participants’ somatotypes were also calculated using the Heath–Carter method as well as body mass index (BMI), which was constructed using the lambda, mu, sigma (LMS) method. Body composition via a bioelectric impedance analysis was also analyzed. The level of special fitness of athletes was determined using tests from the Table Tennis Specific Battery Test, assessing reaction and displacement speeds. Mesomorphic (4.1) and ectomorphic (3.8) profiles were registered for boys and girls, respectively. Boys achieved higher scores than girls for almost all variables, with the exception of ectomorphic somatotype (*p* = 0.274), skinfold triceps (*p* = 0.444), and calf skinfold medial (*p* = 0.609). The relationship between the body height, thickness of the skinfolds of the triceps and suprailiac, biceps, and waist circumference and arm span in all three motor tests was observed, simultaneously significantly higher results were obtained by competitors at the specialist stage of training. Knowledge of the somatic and motor characteristics of young athletes can help coaches in creating a specific training program for improved health and performance, taking into consideration the athletes’ biological development, potential, and pre-disposition.

## 1. Introduction

Table tennis is one of the fastest racquet sports in terms of game speed. Hence, many physical factors are important in order to meet the requirements of this activity [1,2]. The most important factor for a successful table tennis player involves their technical skills, but many other factors can also determine good results in this sport discipline, such as body proportions, somatotype, and body composition. The structure of the athletic body can be related to specialized functions that are required for many specific tasks and allows to understand the limitations of such relationships [3,4]. For example, both body shape and size can influence balance; the lengths of upper and lower limbs may provide a mechanical advantage during play and reaching the ball. On the other hand, excess body mass can have a negative effect on speed and endurance, whereas excess body fat can increase body fatigue [5,6]. Such an association between anthropometric measurements and functional characteristics is highly important for scientists and coaches during athletic potential identification [7]. Most research in table tennis has focused on physical fitness and abilities. Little attention has been directed to the relation between anthropometric measures and special motor fitness [8,9,10]. Several authors have studied the relationship between various anthropological variables and body composition for success in table tennis [11,12,13]. However, some research states that, between the ages of 10 and 14 years, no distinct anthropometric player profile exists [6]. However, determining sport persons’ somatic profiles remains an interesting subject for research within the field of racket sports. 

In the scientific literature, there have been isolated studies addressing the topic of somatotype and anthropometric profiles of table tennis players, but these analyses did not indicate one clearly dominant type of somatic construction. Munivrana and Paušić [14] point out that somatic construction often results not only from lifestyle or diet but also from genetic heritage. The authors carried out anthropometric measurements on a group of 62 young Croatian table tennis players aged 10–14 years playing at the national championships level in the context of sporting success.

The results indicated the dominance of the mesomorphic type of body, which was visible in almost half of the subjects. More than a third of those surveyed had an ectomorphic body type, and more than 16% were endomorphic. Based on comprehensive research, it has been concluded that the somatotype of players of this age is not a key factor in achieving success in table tennis but, rather, increases its likelihood. Similar studies were carried out by Carrasco et al. [13] on a group of 63 players (including 38 boys) aged 10–13 years belonging to the Spanish national table tennis team; Chatterjee et al. [5] also examined 29 players (including 14 men) aged 10–20 years in India. In both cases, the analysis indicated an endomorphic–mesomorphic physique type for boys and endomorphic–ectomorphic for girls.

Khasawneh [15] conducted a study in Jordan to measure anthropometric measurements and determined their relationship to the level of static and dynamic equilibrium, showing a correlation between static balance and hip width and between calf circumference and ankle width and dynamic balance. The study involved 24 young table tennis players playing at the national championship level. Söğüt et al. [16] studied anthropometric profiles and the level of selected components of overall fitness (assessing speed, agility, hand strength, or vertical jump) of young Turkish table tennis players in the context of table tennis serving speed.

The results indicated the importance of several anthropometric characteristics in performing a fast service and the correlation between service speed, body weight, and body mass index (BMI). Behdari et al. [17] compared anthropometric profiles and the level of selected general fitness components taken from EUROFIT European Physical Fitness Test trials. Sixteen Iranian table tennis players participating in the national championships took part in the survey. Comparisons of the top five in the ranking with the others pointed to significant differences in the types of somatotypes among players, indicating the dominant mesomorphic–ectomorphic body-building type for players in the top five of the rankings and mesomorphic–endomorphic for the other players. Generally speaking, the aforementioned studies showed that table tennis players have an average body height, relatively low amount of fat tissue, and high aerobic capacity.

However, despite an increasing number of publications, the research has frequently yielded contradictory results, which engenders an animated discussion on somatic factors that determine the sports development of players in table tennis. Therefore, this study aims to determine what somatotype factors influence the level of physical special fitness of young table tennis players. To achieve this goal, our study was twofold, addressing both physical fitness and somatotype and body composition of young top-level table tennis players. The aim of the study was to identify the impact of some anthropometric measurements on special physical fitness among elite junior table tennis players (girls and boys, at different stages of sport training).

## 2. Materials and Methods

### 2.1. Participants

The study included data collected from 87 young Polish table tennis players aged 13.4 ± 1.74 years (43.7% girls with a body mass M = 47.2 ± 8.97 and height M = 158.2 ± 8.99 and 56.3% boys with a body mass M = 55.1 ± 14.15 and height 165.0 ± 11.81) from the teams of two provinces: Dolnośląskie and Wielkopolskie. The group was selected from a larger study group (*n* = 98 players). The final group (*n* = 87) was composed of players, who meet inclusion criteria (listed below).

The participants were born between 2002 and 2007 and trained in table tennis 3–4 times a week at the targeted or specialized sports training stage (according to the National Table Tennis Development Program in Poland guidelines for 2018–2033). The targeted training stage includes players aged 9–12 years, and the specialized training stage includes players aged 13–17 years. The study group was selected arbitrarily using the following criteria: written consent from parents to participate in the research, membership in the province team, current license of the Polish Table Tennis Association, a minimum three-year training period, health conditions, allowing all physical fitness tests to be carried out, and playing style, requiring the use of rackets with a so-called smooth lining (excluding people using rackets with atypical cladding, such as anti-spin cladding, short pin, or long pin, where play is characterized by a different technique than top-spin strokes used in a battery of special tests).

The study was conducted in compliance with the Declaration of Helsinki and was approved by the local ethics committee: The Bioethics Committee of the Karol Marcinkowski Medical University, Poznań, Poland (No. 543/18). All data were analyzed confidentially.

### 2.2. Protocol of the Study

The design of the study is described in Figure 1. The players were familiarized with the procedures and equipment before the tests began. Anthropometric measures were conducted between 7.00 and 9.00 in the morning and participants were asked to refrain from consuming food before baseline measurements. All measurements were taken and data were collected before exercise (TTSBT). Prior to the start of each trial, the athletes performed the same standard “warm-up” including a block of formative exercises (15 min) and a special “warm-up” on the table (20 min) under the supervision of the coach and were instructed on the correct way to perform it. The trials were performed on an official table tennis court, in the usual training area, in the same time slot (17:00 and 19:00).

### 2.3. Anthropometric Measurements

Anthropometric measurements were taken, according to standard procedures, following the guidelines described by Martin and Saller [18]. All measurements were evaluated by trained personnel. The following characteristics were measured: body height (cm), body weight (kg), sitting height (cm), humerus breadth and femur breadth (cm), five skinfold thicknesses—triceps (cm), suprailiac (cm), subscapular (cm), medial calf skinfold, and calf (cm), the circumferences of the arm relaxed, waist, hip, thigh, and lower leg. Moreover, the arm span of each athlete was measured. Stature (body height) was measured to the nearest 0.1 cm with a stadiometer (GPM, Zurich, Switzerland), with the child standing upright. When sitting height was measured, the subject was asked to sit on a chair with their back and buttocks touching the backboard of the stadiometer, knees directed straight ahead, arms and hands resting at their side. During arm span measurement, which is the maximum distance between the extended middle fingers of the right and left hands, the athletes stood with their backs to the wall with the arms held exactly horizontally at shoulder level. The distance between fingers was measured with a tape measure. Humerus breadth and femur breadth were measured with a sliding caliper (GPM, Zurich, Switzerland). All skinfolds were measured to the nearest 0.1 mm, using a Harpenden caliper (Baty International, Burgess Hill, UK), whereas all circumferences were determined with anthropometric tape (cm).

Body mass was measured with a Tanita MC-780 MA analyzer (Tanita Corp., Tokyo, Japan) with GMON software (version 3.2.8, Tanita Europe BV, Amsterdam, The Netherlands), following the directions and procedures of the manufacturer. All participants were instructed to refrain from exercising and eating or drinking anything for 3 h prior to testing and to void their bladders in order to ensure that the test results were not influenced by body temperature, breathing rate, and/or the presence of food/beverages in the gastrointestinal tract [19]. During the test, children stood erect with their bare feet on the contact electrodes while holding the electrodes of the BIA unit in their hands. The chronological age was calculated to the decimal using the date of birth (day, month, year) and the date (day, month, year) the anthropometric measurements were taken. Decimal age categories were based on one-year intervals (e.g., 12.50 to 13.49 = 13 years). The biological age of body height and body mass of young tennis players was estimated. Growth references for height, weight (body mass), and BMI were constructed with the lambda, mu, sigma (LMS) method, using data from a large, recent, population-representative sample of school-aged children and adolescents in Poland [20]. Participants were classified as underweight, normal weight, or overweight, according to age- and gender-specific cut-off points [21]. All anthropometric dimensions were measured on two occasions for a sample of 87 players to calculate intra-observer technical errors of measurement.

To determine the table tennis players’ somatotypes, the most frequently used model the Heath and Carter method [22] was employed, which was developed on the basis of Sheldon’s type classification. All morphological traits were used to calculate somatotype components of endomorphy (a high degree of subcutaneous fat tissue), mesomorphy (a highly developed musculoskeletal system), and ectomorphy (the linearity of the body is emphasized). In the next step, all participants were assigned to three groups according to the highest value of their somatotype components. A component lower than 2.5 is considered to be low, from 3.0 to 5.0 medium, and from 5.5 to 7.0 high. Values higher than 7.5 are considered extreme [23]. The data obtained using anthropometric measurements were entered into the “Somatotype—calculation and analysis” computer program [24]. This computer program recognizes different combinations of the individual components of a somatotype and places subjects in different groups and sub-groups depending on the influence of each component. All measurements were performed in the morning.

### 2.4. Special Motor Fitness

To determine the level of special fitness of young table tennis players, tests from the Table Tennis Specific Battery Test (TTSBT) were used, assessing reaction and displacement speeds [25]. In reaction speed tests, balls are thrown at high speed (70 balls/minute) to different areas of the table tennis table, and the player has to perform forehand (T1) or backhand (T2) top-spins. The return of balls that touch the table for over 15 s is considered successful. In T3, balls are thrown at high speed (80 balls/minute) to the sides of the table, and the player should alternately perform forehand and backhand top-spins. The return of balls that touch the table for over 15 s is considered successful. The selected options included the use of two strokes, considered basic at the stage of targeted and specialized training, and top-spin forehand and top-spin backhand, previously used for similarly aged participants [26]. The intraclass correlation coefficient (ICC) overall absolute agreement of the assessed parameters of the TTSBT was high (0.85). More specifically, Cronbach’s α (coefficient of reliability or consistency) for T1 was 0.83, for T2 was 0.86, and for T3 was 0.69.

A Tibhar Robo Pro Junior (Racket-Company OHG, Offenbach, Germany), a compact machine with an oscillator and remote control, was used to carry out the test samples. For the test, the machine was programmed to throw balls without rotation and with a central ball spread angle. The balls were thrown in the following variants: (1) top-spin forehand diagonally, (2) top-spin backhand straight, and (3) mixed attempt to play once with top-spin forehand and once with top-spin backhand with the machine in the center of the table. Before the start of each test sample, the competitors performed a block of shaping exercises and special exercises and were informed about the correct way of performing the test samples. All tests were measured on two occasions to calculate the intra-observer technical errors of measurement.

### 2.5. Statistical Analyses

The level of special fitness of young table tennis players was considered a dependent variable in this study. The independent variables were anthropometric measurements, BMI, somatotype categories of trained young male and female table tennis players, their age, gender, and sport training stage. Descriptive statistics were presented as mean and standard deviation or range (minimum–maximum). The difference in mean results between two independent groups was checked using a Student’s *t*-test for independent samples. The normality of the distributions was determined using a Kolmogorov–Smirnov test. In the event of failure to meet the Student’s *t*-test assumption about the normality of the distribution of the studied variable, and in the case of ordinal variables, a Mann–Whitney U test (*Z*) was used to check the significance of differences. Correlations between variables were checked using Spearman’s rank correlation coefficient (r_s_). To accomplish the study goal, separate hierarchical multiple regression models were built, with T1, T2, and T3 as dependent variables. All three models included only the independent variables that were shown to be significantly associated with special motor fitness tests by univariate analyses. In all three models, potential predictors were anthropometric measurements, BMI (according to LMS), somatotype, age, and training experience of the young table tennis players. The threshold of statistical significance for the inclusion of independent variables in the multiple regression models was set at *p* < 0.05. IBM SPSS Statistics 23 (IBM Corp., Armonk, New York, NY, USA) was used for the statistical analyses.

## 3. Results

The sample characteristics and descriptive results are presented in Table 1.

Analysis data from Table 2 connecting the coefficient of variation of the anthropometric characteristics showed that the group was, on average, differentiated in terms of variables related to morphological structure (mean variability 21.2%). In the area analyzed, only the values of the four coefficients of variation (body height, sitting height, humerus diameter, femur diameter) did not exceed 10%. Statistical analysis showed that boys achieved higher scores than girls for almost all variables studied, with the exception of ectomorphic somatotype (*p* = 0.274), skinfold triceps (*p* = 0.444), and calf skinfold medial (*p* = 0.609). The detailed analysis of the data showed that the average BMI (according to LMS) of the subjects was 102.6 (±15.18) points. Nearly 55% of the players surveyed had the correct degree of nutrition, whereas 14% were diagnosed with obesity. Mean somatotypes for table tennis players according to sex and components (endomorphy, mesomorphy, ectomorphy) were presented in Figure 2.

Statistically significant correlations between age and body mass, body height, sitting height, subscapular skinfold girths: biceps (relaxed), waist, hip, calf, and arm span were demonstrated. In addition, age also correlated with humerus and femur diameter measurements. In all cases, the correlations were positive, with older respondents achieving higher values for the above-mentioned measurements (Table 2). A correlating analysis did not show statistically significant links between the players’ ages and their body type.

The group of table tennis players moderately differentiated the area of special motor fitness. Table 3 presents the results obtained by young table tennis players in three special fitness trials.

Statistical analysis did not show significant differences in the results of TTSBT trials between boys and girls. However, it showed that, in all test trials, significantly higher results were obtained by competitors at the specialist stage of training. Results in T1, T2, and T3 showed a moderate volatility strength of less than 30%. In contrast, the correlated analysis demonstrated that older respondents achieved significantly higher results in T1 (*Z* = 0.36, *p* = 0.001), T2 (*Z* = 0.42, *p* < 0.001, and T3 (*Z* = 0. 39, *p* < 0.001). It was also found that respondents with longer training experience achieved higher scores in T1 (*Z* = 0.25, *p* = 0.022) and T2 (*Z* = 0.25, *p* = 0.018). No statistically significant links between training experience and T3 results were demonstrated.

The analyses also showed that young players with high values such as body height (r_s_= 0.28, *p* = 0.009), sitting height (r_s_ = 0.32, *p* = 0.003), and longer arm span (r_s_ = 0.27, *p* = 0.011) performed better in T1. In contrast, lower T1 scores were achieved by subjects with lower results in measuring the thickness of the skin-fat folds of the triceps’ skinfold (r_s_ = −0.29, *p* = 0.006). Higher scores in T2 were achieved by players with high scores of variables such as body height (r_s_ = 0.24, *p* = 0.023), sitting height (r_s_ = 0.33, *p* = 0.002), and with a longer arm span (r_s_ = 0.26, *p* = 0.015). Lower T2 scores were recorded by subjects with lower values in measuring the thickness of the skin-fat folds of the triceps’ skinfold (r_s_ = −0.24, *p* = 0.024) and medial calf skinfold (r_s_ = −0.26, *p* = 0.017). In addition, in the case of T3, higher scores were achieved by tennis players with high scores of such variables as body mass (r_s_ = 0.21, *p* = 0.047), body height (r_s_ = 0.34, *p* = 0.001), sitting height (r_s_ = 0.33, *p* = 0.002), and with a longer arm span (r_s_ = 0.29, *p* = 0.007). The correlation analysis did not show any more statistically significant links between the special motor fitness area and anthropometric measurements. In addition, the correlation analysis did not show statistically significant links between the area of special motor fitness and body type.

To indicate the relationship between the variables tested, a linear regression analysis was performed using the reverse elimination method (Table 4). The first model (for T1) contained five predictors (triceps skinfold, biceps girth (relaxed), waist girth, arm span, endomorphic somatotype). The model explained 28% of the variance of the dependent variable (adjusted R^2^ = 0.283). It was well matched to the data—better than the average and predicted a dependent variable F(5; 81) = 6.38; *p* < 0.001.

The second model (for T2) contained only two predictors (age and triceps’ skinfold). The model explained 21% of the variance of the dependent variable (adjusted R^2^ = 0.209). It was well matched to the data—better than the average and predicted a dependent variable F(2; 84) = 11.06; *p* < 0.001. In the third model (for T3) there were three predictors (age, suprailiac skinfold, endomorphic somatotype). The model explained 24% of the variance of the dependent variable (adjusted R^2^ = 0.235). It was well matched to the data—better than the average and predicted a dependent variable F(4; 82) = 6.29; *p* < 0.001. Hierarchical multiple regression with T1, T2, and T3 as dependent variables showed that almost a quarter of the amount of variance was explained. No other statistically significant associations were observed.

## 4. Discussion

The present study aimed to determine which somatotype factors influence the physical special fitness of young Polish table tennis players. The main results showed that: (i) the examined young table tennis players presented differences in anthropometry and somatotype according to sex and sport training stages; (ii) somatotype was predominantly mesomorphic in boys and ectomorphic in girls; (iii) changes in morphological and special motor fitness characteristics were influenced by age; (iv) links were revealed between the special motor fitness area and selected anthropometric measurements.

It is also important to remember that the morpho-functional characteristics of young athletes are influenced by growth and maturity [11,27], and this fact can offer advantages or disadvantages during competition. The calculated biological age for body height was 91.8% (*n* = 45) in boys and 79.9% (*n* = 30) in girls in accordance with current physical development standards for Polish children and adolescents [20]. For biological age for body mass, an identical relationship was achieved for boys, whereas for girls, it was 82.2% (*n* = 32) of compliance.

The main results showed that the examined young table tennis players were, on average, diversified in terms of variables related to the morphological structure. These results agree with previous studies. The results of other studies conducted on a similar age group mostly indicate a dominant mesomorphic or mesophical–endomorphic physique type [5,13,14]. Differing values in body type, as well as the absence of one clearly dominant body type, indicate that in table tennis at an early stage of sports training, players of different types of somatic construction are trained. In general, the dominant mesomorphic type of physique, particularly among boys, can be explained by the fact that its wide shoulders, long torso, strongly developed and relatively wide hips, and strongly developed, but proportional, lower and upper limbs help young players to manifest themselves in table tennis, which requires a high level of coordination and speed [6,28,29].

However, a gender gap in results can be observed. The mean body-building type index among the boys studied for the ectomorphic type was 3.4, for the mesophical type 4.1, and for the endomorphic type 3.5. In contrast, among girls, the mean body-building index for the ectomorphic type was 3.8, for the mesomorphic type 3.0, and for the endomorphic type 3.3. This result concurs with previous findings e.g., [5,13], which seems to indicate a potential advantage of this types of body constitution in high-level young table tennis players. 

In modern elite table tennis, the appropriate level of a player’s technical skills is extremely important. Therefore, coaches place emphasis on this factor from the moment a young player begins playing table tennis as they strive to reach the elite level. Technical skills are considered a classic limitation in early development, and a period of targeted and specialized stages of sport training is an important time frame for high-potential young athletes to develop their technical skills as a fundament to be able to reach the elite level [6]. 

The correlation analysis showed statistically significant links between the area of special motor fitness and the selected anthropometric measurements, as confirmed in other studies conducted in this age group (e.g., [15] (Jordan players); [12] (Brazilian players)). Based on the results of our studies, it can be concluded that in table tennis, the level of special motor fitness is not determined by a single, dominant somatotype; hence, it is most likely not a key success factor at this level of athletic sophistication. Therefore, it is not possible to discuss the model type of the somatotype of the tested table tennis players. The predominance of the mesomorphic component, or in some cases ectomorphy, reveals the potential advantage of these types of construction, without, however, directly translating into the results of the test trials.

As Pradas et al. [6] noted, this predominance could be explained by the potential biomechanical and technical–tactical advantage of taller players with longer arm span, being able to return a higher number of balls (i.e., covering a wider playing area around the table) and most likely able to produce more force when hitting the ball. Carrasco et al. [13] indicated that although the influence of anthropometric characteristics on young table tennis players is not clear yet, the mesomorphic predominance could play a highly important role in this sport. Investigations carried out using different athletes of the same age reported similar results (e.g., [30]).

Our study on young tennis table players indicated a positive correlation between special motor fitness and the independent variables. However, in the three regression models examined, these correlations explained only a part of the variance. Table tennis is a particularly fast-paced sport that has a small margin of error due to the small size of the ball and racket in relation to the limited playing surface. Thus, to ensure a high level of ball velocity during strokes, correct upper limb muscle mass and arm reach are necessary. Our study showed a relationship between the body height, thickness of the skinfolds of the triceps and suprailiac, biceps, and waist circumference and arm span in the T1, T2, and T3 tests. As Robertson et al. [4] noted, taller players with longer arms can generate more power during the game. Furthermore, table tennis players require considerable upper-limb muscle activity to be able to perform brief explosive movements, change direction rapidly, and effectively hit the ball during decisive strokes such as forehand smashes and forehand top-spin strokes on repeated occasions during the game. In turn, high body weight values would impair performance by speed while possibly increasing the risk of injury. These associations are consistent with the results of other researchers (e.g., [4,6]). Clearly, there are other factors that might mediate these relationships, such as ranking positions, performance level, school environment, differences in maturation, sporting traditions of the local community or a particular family, individual interests, and engagement in sports [4,6,31,32].

In this study, we only focused on anthropometry and special motor fitness, which are important for excelling in racquet sports, particularly table tennis. The main limitation of this study that should be highlighted is that it only assessed young Polish table tennis players, and further studies should be performed with athletes of different nationalities.

## 5. Conclusions

The results presented an idea of an under-explored way of combining important variables in the context of child and youth sport, namely an assessment of the level of motor fitness and anthropometric profile. Somatotype was predominantly mesomorphic in the examined boys and ectomorphic in girls. Higher muscle mass in the upper limbs appeared to be associated with better special motor fitness in young table tennis players. These results could suggest the importance of muscle mass and body height in table tennis players during formative stages.

Knowledge of the somatic and motor characteristics of young table tennis athletes can help coaches in creating specific training programs for optimal physical preparation to improve health and performance, taking into consideration the athletes’ biological development, potential, and predisposition. These results could suggest the importance of anthropometric profile in young table tennis players, particularly during the selection stage; hence, it is worth paying more attention to the physical characteristics because, as studies have shown, they determine a better level of technical preparation of young table tennis players.

## Figures and Tables

**Figure 1 ijerph-18-05279-f001:**
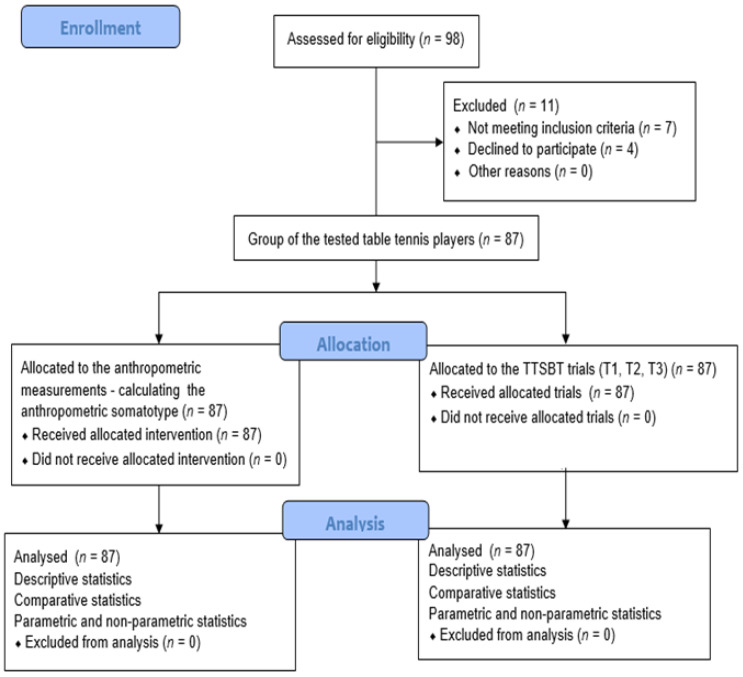
Graphical overview of the study protocol.

**Figure 2 ijerph-18-05279-f002:**
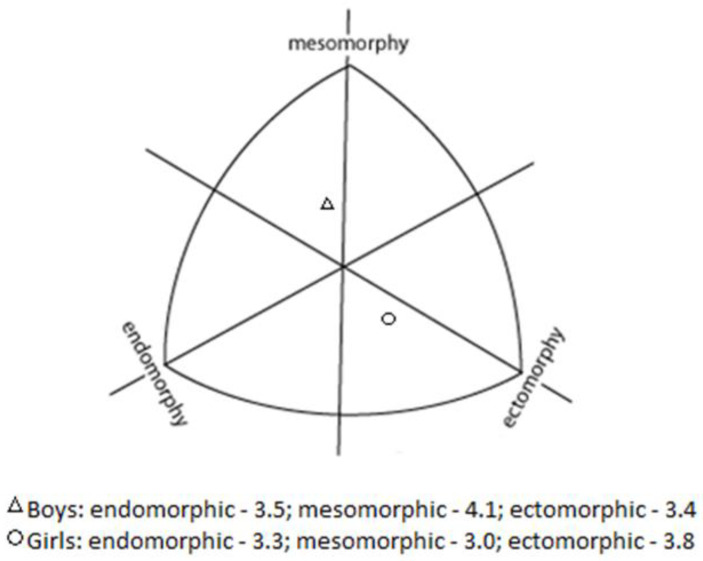
Somatochart—the mean somatotype for boys and girls.

**Table 1 ijerph-18-05279-t001:** Demographic characteristics of the players.

Variable	Girls (*n* = 38, 43.7%)		Boys (*n* = 49, 56.3%)	
	M ± SD	Min–Max	M ± SD	Min–Max
chronological age (years)	13.4 ± 1.76	10.7–16.5	13.7 ± 1.75	11.0–17.0
biological age (years)	12.3 ± 1.52	9.9–16.1	13.4 ± 2.01	8.6–17.9
targeted stage of sport training (*n*, %)	23, 60.5%		27, 55.1%	
specialized stage of sport training (*n*, %)	15, 39.5%		22, 44.9%	

M = mean; SD = standard deviation; Min = minimum value; Max = maximum value.

**Table 2 ijerph-18-05279-t002:** Morphological characteristics of high-level table tennis players for gender, age and stage of sport training.

Variable	Boys (*n* = 49, 56.3%)	Girls (*n* = 38, 43.7%)	Test t Student or U		V (%)	Age		Targeted Stage	Specialized Stage	Test t Student or U	
	M ± SD	M ± SD	t/*Z*	*p*		r_s_	*p*	M ± SD	M ± SD	t/*Z*	*p*
endomorphic somatotype	3.5 ± 1.53	3.3 ± 1.04	0.75	0.452	38.6	−0.03	0.790	3.5 ± 1.35	3.3 ± 1.32	0.72	0.476
mesomorphic somatotype	4.1 ± 1.39	3.0 ± 1.00	−3.95 *	<0.001	36.7	−0.13	0.233	3.8 ± 1.45	3.5 ± 1.17	−0.95	0.342
ectomorphic somatotype	3.4 ± 1.60	3.8 ± 1.27	−1.10	0.274	41.0	−0.03	0.796	3.5 ± 1.44	3.6 ± 1.52	−0.26	0.799
body weight (mass) (kg)	55.1 ± 14.15	47.2 ± 8.97	3.02 *	0.003	24.7	0.55 *	<0.001	47.1 ± 11.61	57.8 ± 11.66	−4.24 *	<0.001
body height (stature) (cm)	165.0 ± 11.81	158.8 ± 8.99	2.70 *	0.008	6.8	0.67 *	<0.001	157.3 ± 9.87	169.0 ± 8.86	−5.71 *	<0.001
sitting height (cm)	85.2 ± 6.72	83.1 ± 5.30	1.57	0.121	7.3	0.70 *	<0.001	81.3 ± 5.40	88.4 ± 4.71	−6.40 *	<0.001
humerus diameter (cm)	6.7 ± 0.51	6.2 ± 0.32	−5.14 *	<0.001	8.1	0.36 *	0.001	6.4 ± 0.53	6.7 ± 0.47	−2.51 *	0.012
femur diameter (cm)	9.5 ± 0.57	8.6 ± 0.48	−6.11 *	<0.001	7.6	0.22 *	0.037	9.0 ± 0.76	9.2 ± 0.59	−1.14	0.254
subscapular skinfold (mm)	9.6 ± 4.95	9.4 ± 3.93	−0.41	0.681	47.4	0.26 *	0.016	9.3 ± 4.74	9.8 ± 4.22	−1.46	0.144
triceps skinfold (mm)	12.2 ± 5.01	13.0 ± 3.58	−0.77	0.444	35.3	−0.06	0.557	13.2 ± 4.62	11.6 ± 4.04	1.66	0.100
suprailiac skinfold (mm)	12.7 ± 8.02	10.7 ± 4.21	−0.52	0.602	56.3	0.07	0.497	11.7 ± 6.13	12.0 ± 7.43	−0.12	0.904
calf skinfold (mm)	13.5 ± 5.94	12.7 ± 4.76	0.66	0.509	41.3	−0.16	0.134	14.3 ± 5.72	11.6 ± 4.65	2.39 *	0.019
medial calf skinfold (mm)	14.2 ± 6.74	14.9 ± 4.72	−0.51	0.609	40.8	−0.19	0.080	15.5 ± 5.67	13.1 ± 6.04	1.92	0.058
biceps arm girth-relaxed (cm)	25.2 ± 3.58	23.5 ± 2.36	2.53 *	0.013	13.1	0.48 *	<0.001	23.4 ± 3.24	25.8 ± 2.57	−3.81 *	<0.001
waist girth (cm)	68.8 ± 8.03	61.8 ± 5.12	−4.23 *	<0.001	11.7	0.39 *	<0.001	64.0 ± 7.50	68.1 ± 7.45	−2.57 *	0.010
hip girth (cm)	84.9 ± 9.40	81.8 ± 7.12	1.69	0.095	10.3	0.51 *	<0.001	80.6 ± 8.21	87.4 ± 7.52	−3.95 *	<0.001
thigh girth (cm)	49.2 ± 6.11	48.7 ± 5.13	0.36	0.719	11.6	0.46 *	<0.001	47.2 ± 5.55	51.4 ± 4.95	−3.41 *	0.001
calf girth (cm)	32.9 ± 3.47	31.0 ± 3.01	2.69 *	0.008	10.6	0.38 *	<0.001	31.3 ± 3.65	33.1 ± 2.72	−2.60 *	0.011
arm span (cm)	162.5 ± 26.84	156.6 ± 10.62	−2.84 *	0.005	13.4	0.60 *	<0.001	153.4 ± 24.35	168.7 ± 12.38	−4.46 *	<0.001
BMI (body mass index) (LSM)	106.2 ± 16.15	98.0 ± 12.61	−2.66 *	0.008	14.8	−0.01	0.927	104.4 ± 15.75	100.2 ± 14.22	−1.16	0.246

r_s_—Spearman rank correlation coefficient; M = mean, SD = standard deviation; t = Student *t*-test value; * *Z* = Mann−Whitney U test value; *p* = level of significance; V = coefficient of variation.

**Table 3 ijerph-18-05279-t003:** Results of the Table Tennis Specific Battery Test (TTSBT) (the number of balls that touch the table in 15 s): comparison by gender and training stages.

Variable	Boys	Girls	U Manna-Whitneya Test	
	M ± SD	M ± SD	*Z*	*p*
Test 1	13.4 ± 3.32	13.6 ± 3.08	−0.34	0.734
Test 2	13.1 ± 3.91	13.4 ± 3.17	−0.40	0.686
Test 3	10.9 ± 3.03	10.4 ± 3.05	−0.68	0.496
	**Targeted Stage**	**Specialized Stage**	***Z***	***p***
Test 1	12.5 ± 3.16	14.8 ± 2.77	−3.28 *	0.001
Test 2	12.0 ± 3.52	14.9 ± 2.98	−3.96 *	<0.001
Test 3	9.8 ± 3.09	11.9 ± 2.54	−3.47 *	0.001

M—mean, SD—standard deviation, * *Z*—Mann-Whitney U test value, *p*—level of significance.

**Table 4 ijerph-18-05279-t004:** Regression analysis factors for T1, T2 and T3.

	B	SE	β	T	*p*
**Model for T1:**					
(constant)	11.66	2.91		4.01	0.000
triceps skinfold [cm]	−0.62	0.14	−0.858	−4.47	0.000
biceps girth (relaxed) [cm]	0.62	0.22	0.616	2.79	0.007
waist girth [cm]	−0.29	0.10	−0.699	−2.80	0.006
arm span [cm]	0.06	0.02	0.368	2.71	0.008
endomorphic somatotype	1.40	0.56	0.582	2.51	0.014
**Model for T2:**					
(constant)	5.08	2.98		1.70	0.092
age (in years)	0.78	0.20	0.378	3.87	0.000
triceps skinfold [cm]	−0.18	0.08	−0.221	−2.26	0.026
**Model for T3:**					
(constant)	−1.27	4.54		−0.28	0.781
age (in years)	0.52	0.17	0.298	3.01	0.003
suprailiac skinfold [mm]	0.29	0.10	0.635	2.94	0.004
endomorphic somatotype	−1.22	0.49	−0.534	−2.46	0.016

B—the unstandardized beta, SE, B—the standard error for the unstandardized beta, β—the standardized beta, t—the *t* test statistic, *p*—the probability value.

## Data Availability

The data presented in this study are available on request from the corresponding author. The data are not publicly available due to no access to publicly accessible repository.

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
