# Peer review of "Somatic Characteristics and Special Motor Fitness of Young Top-Level Polish Table Tennis Players"

_ijerph, 2021, doi:10.3390/ijerph18105279_

Round 1

Reviewer 1 Report

As per annexed considerations

Author Response

Dear Reviewer,

In responses to yours comments in the review of the manuscript " Somatic characteristics and special motor fitness of young top-level Polish table tennis players", we would like to inform you about the undertaken corrections and completions.

We appreciate your thoughtful comments and hope that this revision will be considered much stronger as a result.  The authors’ responses are given separately to each Reviewer’s comment. We have tried to address your concerns as fully as possible.  Below are the changes that we have introduced.

Thank you for your consideration and we look forward to hearing from you soon.

Reviewer 1.

Authors: First of all, we would like to express our gratitude to Reviewer 1 for the time taken to review our paper and for providing us with comments/suggestions helpful to improve the quality of this paper. We would like to thank for the Reviewer’s valuable contributions. We have found the Reviewer’s criticism and recommendations positive and very constructive. The changes to the paper have been made using red font in the new version of the paper.

General comments:

Title Are presented satisfactorily.

Abstract It is written in an unstructured form, but the methodology is written in a summarized form  and  numerical  results  have  not  been  presented,  which  makes  it  difficult  to understand the manuscript in more depth. It is suggested that the main numerical data of the study be inserted in the abstract.

Response: The Reviewer’s proposal was taken into account.

Introduction Are presented satisfactorily.

Response: Thank you for a positive assessment of this part of our paper.

Methods It  should  present  more  clearly  the  design  of  the  study.  A  CONSORT  should  be presented in order to get a better view of the study design.

Response: We would like to thank the Reviewer for their valuable remark. Flow diagram of the progress through the phases of the randomised trial of the group of table tennis players. The figure illustrated the  design  of  the  study was added.

The item participants are well described. In the item Anthropometric Measurements, the instruments must be better described, with manufacturer, city and country.

Response: Thank you for Your recommendation, we have described all instruments in more details.

Furthermore, there is a reference (Toth et al., 2014), which is not in the magazine format.

Response: Thank you for your note. We actually overlooked this one reference, so it was added to the final list of references.

With regard to Special Motor Fitness, the presentation of the instruments used in the research should be reviewed. In the statistical analysis in some tests, parametric and nonparametric assessments are mentioned and in others only nonparametric assessments are mentioned. Please comment.  On  the  other  hand,  some  statistical  procedures  should  be  further developed, I suggest consulting Cohen (1988).

Response: Thank you for all valuable comments and information. In the opinion of the authors, section Special motor fitness presented an in-depth description of the test trials. A detailed description of the Table Tennis Specific Battery Test (TTSBT) is included in the positions cited in the manuscript: Gomes F., Amaral F., Venture A., & Agular, J. (2000). Table Tennis specific test battery, International Journal of Table Tennis Sciences, 4, 11-18; Katsikadelis, M., Pilianidis, T., & Mantzouranis, N. (2014). Test-retest reliability of the “table tennis specific battery test” in competitive level young players. European Psychomotricity Journal, 6(1), 3-11.

Parametric and non-parametric methods were used in the analysis of the study results, depending on the fulfillment of the assumption of normal distribution of the studied variables. Parametric tests require the assumption that the sample is drawn from a population with a normal distribution. For non-parametric tests, no such assumption is required. Analysis of the Kolmogorov-Smirnov test showed that the following variables had distributions inconsistent with the normal distribution: mesomorphic somatotype, humerus diameter, femur diameter, subscapular skinfold, waist girth, hip girth, thigh girth, arm span and for BMI (LSM). Also, the distributions of the variables T1, T2 and T3 were not close to normal distribution.

Results Are presented satisfactorily. However, I suggest you look at the questions of the methodology.

Response: Thank you for a positive assessment of this part of our paper.

Discussion Are presented satisfactorily.

Response: Thank you for a positive assessment of this part of our paper.

Conclusion Are presented satisfactorily.

Response: Thank you for a positive assessment of this part of our paper.

References Of the 31 references, 17 have been published for more than five years. It would be important to update and revise the formatting in order to comply with the magazine's rules.

Response: The changes were made as suggested by the Reviewer. All references have been checked in the text to meet the journal’s requirements.

Overview The manuscript presented addresses a relevant research topic. It would be advisable to do a general review.

Reviewer 2 Report

Dear authors,

just completed reviewing your works, first of all, all of the contents of your manuscript are well organized and followed potential protocols of scientific research within the domain of sports science. This study was desinged to verify the level of motor fitness relating to anthropometric profiles of youth players. Your study potentially help audiences to understand young athletes biological development, optimal physical preparation and their performance development. However, I feel there are still some missing parts in terms of the oraganization of the results. Please consider the followings:

  1. instead of presention statistical results of your findings in table format. it would be better for potential audiences if you add figures and graphs for your findings.
  2. also please add a figure depicint your procedural detailes (data collection process or steps of it)

I believe the comments above help your study to further develop

Author Response

Dear Reviewer,

In responses to yours comments in the review of the manuscript " Somatic characteristics and special motor fitness of young top-level Polish table tennis players", we would like to inform you about the undertaken corrections and completions.

We appreciate your thoughtful comments and hope that this revision will be considered much stronger as a result.  The authors’ responses are given separately to each Reviewer’s comment. We have tried to address your concerns as fully as possible.  Below are the changes that we have introduced.

Thank you for your consideration and we look forward to hearing from you soon.

Authors: First of all, we would like to express our gratitude to Reviewer 2 for the time taken to review our paper and for providing us with comments/suggestions helpful to improve the quality of this paper. We would like to thank for the Reviewer’s valuable contributions. We have found the Reviewer’s criticism and recommendations positive and very constructive. The changes to the paper have been made using red font in the new version of the paper.

Dear authors, just completed reviewing your works, first of all, all of the contents of your manuscript are well organized and followed potential protocols of scientific research within the domain of sports science. This study was desinged to verify the level of motor fitness relating to anthropometric profiles of youth players. Your study potentially help audiences to understand young athletes biological development, optimal physical preparation and their performance development.

Thank you for a positive assessment of the paper. We appreciate the positive feedback from the Reviewer.

However, I feel there are still some missing parts in terms of the oraganization of the results. Please consider the followings:

  • instead of presention statistical results of your findings in table format. it would be better for potential audiences if you add figures and graphs for your findings.

Response: The obtained results were presented in the manuscript in the form of statistical tables, which made it possible to systematically organize the numerical data in the adopted way. Thanks to their content, the tables facilitated a better understanding of the studied issue and enabled a comparison of the variables included in them. The use of tables were therefore a question of their functionality. An additional advantage is the precision in presenting the individual values of the studied variables, which would be difficult to achieve on a graphs. Therefore, tables seem to be, in the authors' opinion, useful and more informative for reporting the research results obtained.

  • also please add a figure depicint your procedural detailes (data collection process or steps of it)

Response: We would like to thank the Reviewer for their valuable remark. Flow diagram of the progress through the phases of the randomised trial of the group of table tennis players. The figure illustrated the  design  of  the  study was added.

I believe the comments above help your study to further develop.

Reviewer 3 Report

This paper analyzes anthropometric characteristics and specific table tennis test on an adolescent sample. In general, the article is clear and well written. However, there are some concerns that should be addressed.

Line 10. What do authors mean by special?

Line 12. "Anthropometric measurements" would sound better

Line 13. Elbow and knee width are not anthropometric measurements. The official terms must be used. Same for line 119

The introduction is well written and complete. I do not think it is necessary to do any changes on it.

120-121. You mentioned two calf skinfolds. Does one of them mean to me the thigh skinfold?

In my opinion, the discussion is a little poor and authors should deepen a little more in it.

Author Response

Dear Reviewer,

In responses to yours comments in the review of the manuscript " Somatic characteristics and special motor fitness of young top-level Polish table tennis players", we would like to inform you about the undertaken corrections and completions.

We appreciate your thoughtful comments and hope that this revision will be considered much stronger as a result.  The authors’ responses are given separately to each Reviewer’s comment. We have tried to address your concerns as fully as possible.  Below are the changes that we have introduced.

Thank you for your consideration and we look forward to hearing from you soon.

Authors: First of all, we would like to express our gratitude to Reviewer 3 for the time taken to review our paper and for providing us with comments/suggestions helpful to improve the quality of this paper. We would like to thank for the Reviewer’s valuable contributions. We have found the Reviewer’s criticism and recommendations positive and very constructive. The changes to the paper have been made using red font in the new version of the paper.

This paper analyzes anthropometric characteristics and specific table tennis test on an adolescent sample. In general, the article is clear and well written. However, there are some concerns that should be addressed.

Line 10. What do authors mean by special?

Response: Special physical fitness consists in the advancement of technical skills and abilities in specific disciplines or sports competitions. In this case in table tennis. In turn, physical fitness can be defined as the current ability to perform all motor activities.

Line 12. "Anthropometric measurements" would sound better

Response: The Reviewer’s proposal was taken into account - modified accordingly.       

Line 13. Elbow and knee width are not anthropometric measurements. The official terms must be used. Same for line 119

Response: We would like to thank the Reviewer for their valuable remark. We have used the correct nomenclature throughout the work.

The introduction is well written and complete. I do not think it is necessary to do any changes on it.

Response: Thank you for a positive assessment of this part of our paper.

120-121. You mentioned two calf skinfolds. Does one of them mean to me the thigh skinfold?

Response: We actually measured two calf skinfold: calf skinfold (a vertical pinch parallel to the long axis of the leg) and medial calf skinfold (a point on the medial (inside) surface of the calf, at the level of the largest circumference).

In my opinion, the discussion is a little poor and authors should deepen a little more in it.

Response: The first paragraph of the discussion was firstly structured with the main aim of the study. Then, we focused on distinguishing the research goal and the most important results. Information irrelevant to the work has been removed from the text.